# OpenReview forum: "Get RICH or Die Scaling: Profitably Trading Inference Compute for Robustness"
_NeurIPS.cc/2025/Conference — Submitted to NeurIPS 2025_

### Official Review · Reviewer_oVEA · 2025-06-27

**Clarity:** 1
**Significance:** 4
**Originality:** 4
**Rating:** 5
**Confidence:** 3

**Summary:**

+1 to the deep cut 50 cent reference

I am very intrigued by this paper. The Zaremba paper sparked my intense interest earlier in the year because inference time compute is a critical trend in AI right now as we hit the “data-wall”. The intense interest in reasoning time compute provokes a natural and important question; how does inference time compute interact with adversarial robustness? We have had ~1 impactful paper on this topic so far, which is the Zaremba paper.

This paper also helpful adds to this new literature by focusing on multi-modal models  with a new hypothesis, called the RICH hypothesis. I think this hypothesis synthesizes some vibes-based impressions/folklore from previous generations of adversarial robustness research into a relatively testable hypothesis that is quite interesting. OOD adv attack susceptibility not benefiting extended inference time models but robust models -> interpretable attacks that do benefit from extended inference time is an intriguing hypothesis.

My understanding of the main claim of this paper is that researchers need to improve inference time compute methods and base model robustness in parallel? I think it is possible to over-read Zaremba as “inference time compute gives you robustness for free”. In contrast, this paper says the two together are stronger because more robust models -> more interpretable perturbations -> more amenable to reasoning benefits.

**Questions:**

my questions are in the weaknesses section.

**Ethical Concerns:**

["Major Concern: Data quality and representativeness"]

**Final Justification:**

technical edits by the authors remove my confusion about which experiments correspond to which figures, ethical concern fully adressed

**Limitations:**

Critical Issue: No experimental details on o1 results from Figure 1 and apparently missing the claude results which are discussed in multiple sections, especially section 3.2. I am very open to revising my score to acceptance if the authors report the results they claim to have done. Where are the results discussed in lines 191 to 202? Do they not exist? Did I somehow not find them? Please help. Maybe the authors confused sonnet and o1?

This issue damages the credibility of the paper.

**Paper Formatting Concerns:**

see above

**Quality:**

1

**Strengths And Weaknesses:**

Very intriguing paper, but confusing layout/organization.

Critical Issue: No experimental details on o1 results from Figure 1 and apparently missing the claude results which are discussed in multiple sections, especially section 3.2. I am very open to revising my score to acceptance if the authors report the results they claim to have done. Where are the results discussed in lines 191 to 202?? Do they not exist? Did I somehow not find them? Please help. Maybe the authors confused sonnet and o1??

The main technical weakness of the first set of experiments in this  paper is the way they
instantiate “Inference-Time Compute”, which is by repeating the “correct” answer in the Textual input to the vision language model.  The authors remark on this issue at line 175.

Section 3.2 is stronger because it uses more realistic inference time compute techniques like CoT. The main weakness of this section is the limited lack of reproducibility because Claude is closed weight and the extended-thinking mode may be quietly updated/modified by Anthropic. This is an acceptable and necessary tradeoff to test the authors hypothesis on a truly frontier model.

Misc:

When the authors introduce the Attack-Bard results for Chain of Thought and Augmented Reasoning, it is unclear to me where to actually find the results! Please make this more clear? Where are the results?

Fig 3: caption should be more clear about difference between high and low, add more info to make this caption more self contained.

Table 1: Would be good to have this experiment duplicated for more attack images? Why is it so few examples.

Table 2: The very minimal increase in robustness seems to weaken the rich hypothesis substantially. CoT is a much a much more “typical” or standard form of inference time compute.  Am I misunderstanding these results? Please clarify and if so edit this section to demonstrate that this is at best “neutral” evidence for the rich hypothesis.

Q: They focus on gradient based perturbation style attacks. Do you have any theories on how this work may transfer to semantic adversarial attacks? e.g. “visual” prompt injections?

Q: experiments with an open weights reasoning model where we can inspect reasoning traces could have been beneficial. Maybe these were not available at the time this research was conducted. This would provide strong evidence.

Q: Again, where are the Sonnet results? Where are the experimental details for figure 2?

Niggling Complaints.

I don’t like how the methodology section goes into a lot of detail about each experiment and then delays actually showing/discussing the experimental results until the next section. I would radically shorten the methodology section to include only details that are used in each experiment. Then have the introduction/analysis of each experiment be contained.

E.g Merge 3.1 and 4.1! This allows the reader to maintain an ongoing “context” of each experiment.

---

> ### Author Rebuttal · Authors · 2025-07-31
>
> Thank you for your review! We are happy to hear that you are “very open to revising [your] score to acceptance if the authors report the results they claim to have done”. Our revised manuscript addresses the lack of clarity that we believe led to this core concern. As we discuss below, all of the discussed experiments do indeed have their results reported in the submission. We would be happy to discuss any new or remaining concerns during the response period.
>
> > My understanding of the main claim of this paper is that researchers need to improve inference time compute methods and base model robustness in parallel? I think it is possible to over-read Zaremba as “inference time compute gives you robustness for free”. In contrast, this paper says the two together are stronger because more robust models -> more interpretable perturbations -> more amenable to reasoning benefits.
>
> Exactly! Indeed, our findings push back against the risky misconception that inference scaling alone solves robustness. Going beyond Zaremba et al. (2025), we show that increasing model size or inference compute, without addressing representation robustness, offers limited benefit under adversarial conditions. In contrast, using robust base models enables inference-time reasoning to actively defend against attacks, improving performance even in challenging settings.
>
> > Critical Issue: No experimental details on o1 results from Figure 1 and apparently missing the claude results which are discussed in multiple sections, especially section 3.2... Where are the results discussed in lines 191 to 202?... Where are the experimental details for figure 2?
>
> Figure 2 has both o1 and Sonnet results; i.e., “the experimental details for figure 2” are the Sonnet-augmented-reasoning experimental details given on “lines 191 to 202”. Our revision will clarify Figure 2’s o1 and Sonnet experiments, respectively, as follows:
> - *Sonnet:* The Sonnet-augmented-reasoning results in Figure 2 are described by the  experimental details “discussed in lines 191 to 202” and are discussed in Section 2 (e.g. Lines 94 and 98). Our revision will use the extra page to create a new experiment subsection in Section 4 that describes the Sonnet experiment in a self-contained manner. Note that we will also take the reviewer’s advice to discuss methodology and experiment in the same area to allow “the reader to maintain an ongoing ‘context’ of each experiment.” Our revision will also keep Section 2’s minimal discussion of Figure 2’s Sonnet results because we believe that they provide critical background for the relatively new problem area that we explore.
> - *o1:* Unfortunately, Zaremba et al. (2025) tests a proprietary model and does not provide extensive experimental details. Our revision will clarify that the o1 results in Figure 2 are copied from Zaremba et al. (2025) and will include the experimental details they do provide, which are:
>    - class label information is provided within the prompt,
>    - the model predicts the class label given the image,
>    - the attacker is considered successful if the model’s prediction is incorrect.
>
> > The main technical weakness of the first set of experiments in this paper is the way they instantiate “Inference-Time Compute”
>
> Notably, our submission studied both black-box and white-box attacks, and for black-box attacks we instantiated inference compute via both Sonnet-augmented reasoning (Figure 2) and LLaVA CoT (Table 2).
>
> The reviewer is correct that the white-box attacks involved repeating the correct answer in the model’s context. We took this approach to make the PGD attack stronger: with a static context, the PGD attacker is attacking a fixed function – a fixed model context window – that allows it to make consistent progress over a series of PGD steps. The alternative approach would be to risk interrupting the PGD attacker’s progress by letting the model reason freely such that the context window is constantly changed.
>
> To test the effect of letting the model reason freely, Zaremba et al. (2025) used black-box attacks from the Attack-Bard dataset. We also test the effect of letting the model reason freely with the same black-box attacks; however, we also include the aforementioned white-box experiments.
>
> > [limited reproducibility with frontier models]
>
> Our revision will enhance reproducibility by including the specific version of Sonnet that we used (“claude-3-7-sonnet-20250219”), though we agree that exact reproducibility is hard to guarantee with closed-weight models.
>
> Moreover, we now have new and improved results that achieve this “realistic” quality (see below). First, we improved the methodology used in Table 2. Second, we added the requested “visual prompt injection” attack to our white-box methodology.
>
> > Fig 3: caption should be more clear about difference between high and low, add more info to make this caption more self contained.
>
> We will update Figure 3's caption to add details about these Sonnet-augmented-reasoning experiments. Notably, given image descriptions from the various LLaVA models, Figures 2 and 3 used Sonnet without reasoning for the “low” setting, and 16,000 token thinking budget in the “high” setting.
>
> > Table 1: Would be good to have this experiment duplicated for more attack images? Why is it so few examples.
>
> Our revision (and the below table) increases Table 1’s data by a factor of four, such that we now attack models on four different images, with three unique attack targets per image (i.e., 12 attacks per table cell). The original table used fewer examples due to limited compute time. The updated table maintains the trend observed originally, while reducing the standard errors of the estimated PGD steps (i.e. attack difficulties). In particular, Delta2LLaVA receives the clearest benefits from increasing inference compute, and it is the most robust base model.
>
> **Table R1**: PGD steps required for a successful attack across models, perturbation budget ε, and inference-compute levels K. Mean (standard error) computed on three attack variations (color, shape, texture) for the Red Ball, Speed Limit Sign, iPod and Sea Urchin images.
> | ε | K | LLaVA-v1.5 | FARE-LLaVA-v1.5 | Delta2LLaVA-v1.5 |
> |---|---|---|--|---|
> | 16| 0 | 5.7 (0.7) | 18.8 (6.8) | Attack Failed |
> | 16 | 1 | 7.2 (0.7) | 24.7 (6.5) | Attack Failed |
> | 16| 3 | 7.6 (0.7) | 26.5 (7.1) | Attack Failed |
> | 16| 5 | 7.0 (0.6) | 27.2 (7.0) | Attack Failed |
> | 64| 0 | 6.2 (0.8) | 6.7 (1.1) | 25.4 (7.8) |
> | 64| 1 | 7.2 (0.7) | 8.0 (1.2) | 50.8 (9.8) |
> | 64| 3 | 7.6 (0.7) | 9.3 (1.4) | 57.5 (8.9) |
> | 64 | 5 | 7.4 (0.8) | 9.2 (1.4) | 63.2 (8.4) |
>
> > Table 2: The very minimal increase in robustness seems to weaken the rich hypothesis substantially.
>
> Our revision improves Table 2's experimental design and finds statistically significant support for the RICH. Previously, CoT didn't provide a significant improvement on clean data (see the first two rows of Table 2). We altered our CoT prompt to illustrate to the VLM how to format its output, which caused all models to show a benefit from CoT on clean data. Given this baseline improvement from CoT, we then computed the benefit of CoT on adversarial data. Consistent with the RICH, we found that only robust models benefitted from CoT on the adversarial data (at the p-value < 0.01 level, using a McNemar Test), supporting the RICH hypothesis.
>
> **Table R2**: Attack Bard Black Box 30-Way Classification on LLaVA Family
> | Evaluation | Model | No CoT | CoT | CoT Improvement at 0.01 Significance Level? (P-Value) |
> |------------|-------|--------|-----|-------------------------------------------------------|
> | Clean | LLaVA-v1.5 | 69.5 | 82.0 | **Yes** (1.376e-4) |
> | Clean | FARE-LLaVA-v1.5 | 61.5 | 71.0 | **Yes** (9.414e-4) |
> | Clean | Delta2LLaVA-v1.5 | 62.0 | 72.5 | **Yes** (3.919e-3) |
> | Adv | LLaVA-v1.5 | 38.0 | 44.5 | **No** (4.233e-2) |
> | Adv | FARE-LLaVA-v1.5 | 56.0 | 65.5 | **Yes** (4.621e-3) |
> | Adv | Delta2LLaVA-v1.5 | 62.0 | 73.0 | **Yes** (4.509e-3) |
>
> > Do you have any theories on how this work may transfer to semantic adversarial attacks? e.g. “visual” prompt injections?
>
> Great question! Our revision will include the following exciting result that further supports the RICH. We inserted a visual prompt injection into a clean Attack-Bard image, then we performed a white-box PGD attack that encouraged the model to repeat the visually injected text instead of describing the image as asked. When we scaled the inference compute, only robust models benefitted, as shown by the increase in the loss of the PGD attacker below.
>
> **Table R3**: Only robust LLaVA models obtain added robustness (measured with PGD attacker’s loss) from inference compute scaling when faced with visual prompt injections.
> | Model | Base Model Robustness | Inference Compute | Step 100 Attacker Loss (&uarr;) | Step 200 Attacker Loss (&uarr;) | Step 300 Attacker Loss (&uarr;) | Inference Compute Robustness Effect |
> | --- | --- | --- | - | - | - | - |
> | Delta2LLaVA-v1.5 | High | Low | 13.5 (0.0) | 12.7 (0.1) | 12.4 (0.0) | -- |
> | Delta2LLaVA-v1.5 | High | High | 21.2 (0.0) | 21.1 (0.0) | 21.1 (0.0) | Positive |
> | FARE-LLaVA-v1.5 | Medium | Low | 7.5 (0.4) | 7.0 (0.0) | 7.0 (0.5) | -- |
> | FARE-LLaVA-v1.5 | Medium | High | 9.3 (1.1) | 7.4 (0.7) | 7.2 (0.3) | Neutral/Positive |
> | LLaVA-v1.5 | Low | Low | 6.4 (1.4) | 2.6 (2.9) | 2.0 (2.6) | -- |
> | LLaVA-v1.5 | Low | High | 2.9 (0.8) | Attack Success | Attack Success | Negative |
>
> > Q: experiments with an open weights reasoning model where we can inspect reasoning traces could have been beneficial.
>
> We will add to the revision’s appendix the CoTs used by the open weight models we study. Additionally, we will mention the reviewer’s suggestion as a direction for future work.
>
> > have the introduction/analysis of each experiment be contained… E.g Merge 3.1 and 4.1! This allows the reader to maintain an ongoing “context” of each experiment.
>
> Good idea! We will adopt this approach in our revision.

---

> ### Comment · Reviewer_oVEA · 2025-08-04
> **addresses my concerns fully, changed score**
>
> Hello!
>
> Thank you for addressing my concerns, which were mostly about the organization of experimental details. The paper has a bevy of intriguing and thoughtful experiments, which is a key strength of the paper, but I think that makes being explicit about which figures correspond to which experiments extra important. I love the added visual prompt injection experiments.
>
> I believe the author's changes and edits alleviate my concerns and allow the scientific impact of the RICH hypothesis to shine through.
>
> eg, I totally misunderstood that the "o1" experiment was merely reporting the Zaremba et al results and thought that was a novel experiment in this paper.
>
> Accordingly, I have changed my score to 5.

---

> > ### Author Response · Authors · 2025-08-06
> >
> > We are very grateful for your review and subsequent engagement. Your contributions have improved our manuscript's quality and clarity, and we're glad to hear that you "love the added visual prompt injection experiments" -- they were a great idea you had!
> >
> > Thank you,
> >
> > Authors of 24976

---

### Official Review · Reviewer_mxf9 · 2025-06-30

**Clarity:** 3
**Significance:** 3
**Originality:** 3
**Rating:** 5
**Confidence:** 3

**Summary:**

The authors discuss in their paper that robust multi-models get much better at defending against attacks when given more time to "think" during inference, while non-robust models don't improve much. This creates a "rich-get-richer" effect, where already-strong models benefit most from extra reasoning time.

**Questions:**

1. Ln:159: What makes PGD novel in this context?
2. Why TTA was not considered important for this paper?
3. Ln311: While the insights are interesting to me, it is somehow not explained in the paper how close this could be helpful for real-world defenses? (practability)

**Ethical Concerns:**

["NO or VERY MINOR ethics concerns only"]

**Limitations:**

- Number of models in experiment could be larger.

**Quality:**

3

**Strengths And Weaknesses:**

Strengths
 - Findings: Robust models at inference time become more robust.
 - The findings are related to the vision part, which has been the weak spot of multimodal, and therefore, I think that is an important contribution.
 - Improves defense with CoT.

Weaknesses
 - Experiments: it is not clear if there is a difference if the model has a different size.
 - Related work: Test-time adaptation (TTA) has never been discussed. This could be interesting.

---

> ### Author Rebuttal · Authors · 2025-07-31
>
> Thank you for your insightful review! We are happy to hear that you believe our work is an "important contribution". Moreover, thank you for emphasizing that visual robustness has been a weakness for vision language models. Yes, our objective is to help address this weakness by providing a hypothesis, and empirical evidence, on the mechanism for when increased inference compute confers robustness benefits. Our findings push back against the common misconception that scaling alone solves robustness: additional inference compute without representation robustness yields little benefit. Thank you for your concern about the lack of model size diversity. We direct you to our expanded results with frontier open-weight models which bolsters the generality of the Robustness-from-Inference-Compute Hypothesis (RICH).
>
> > Experiments: it is not clear if there is a difference if the model has a different size… Number of models in experiment could be larger.
>
> Our revision will address both of these concerns by adding more models to our experiments, and by choosing new models that are larger than those we explored. As shown in Table R1 below, LLama-3.2-Vision-High-90B and Qwen-2.5-VL-72B-Instruct do not exhibit substantial, statistically significant gains from CoT on attacked data despite doing so on clean data. This supports the RICH, consistent with our submission’s Table 2 and the updated version of Table 2 below (Table R2).
>
> **Table R1** Attack Bard Black Box 1000-way Classification On Large Models
> | Evaluation | Model | No CoT | CoT | CoT Improvement at 0.01 Significance Level? (P-Value)|
> |------------|-------|--------|-----|--------------------------------------------|
> | Clean | LLama-3.2-Vision-High-90B | 63.5 | 68.5 | **No** (1.921e-2) |
> | Clean | Qwen-2.5-VL-72B-Instruct | 57.0 | 67.5 | **Yes** (5.557e-4) |
> | Adversarial | LLama-3.2-Vision-High-90B | 27.0 | 27.5 | **No** (7.905e-1) |
> | Adversarial | Qwen-2.5-VL-72B-Instruct | 13.0 | 18.0 | **No** (1.273e-2) |
>
>
> **Table R2** Attack Bard Black Box 30-Way Classification on LLaVA Family
> | Evaluation | Model | No CoT | CoT | CoT Improvement at 0.01 Significance Level? (P-Value) |
> |------------|-------|--------|-----|-------------------------------------------------------|
> | Clean | LLaVA-v1.5 | 69.5 | 82.0 | **Yes** (1.376e-4) |
> | Clean | FARE-LLaVA-v1.5 | 61.5 | 71.0 | **Yes** (9.414e-4) |
> | Clean | Delta2LLaVA-v1.5 | 62.0 | 72.5 | **Yes** (3.919e-3) |
> | Adversarial | LLaVA-v1.5 | 38.0 | 44.5 | **No** (4.233e-2) |
> | Adversarial | FARE-LLaVA-v1.5 | 56.0 | 65.5 | **Yes** (4.621e-3) |
> | Adversarial | Delta2LLaVA-v1.5 | 62.0 | 73.0 | **Yes** (4.509e-3) |
>
> > Related work: Test-time adaptation (TTA) has never been discussed. This could be interesting… Why TTA was not considered important for this paper?
>
> Thank you for the suggestion! Our submission will discuss TTA as an interesting direction for future work, referencing how TTA can induce adversarial vulnerabilities (Wu et al., 2023). Please let us know if you have any additional suggestions.
>
> > Ln:159: What makes PGD novel in this context?
>
> Great question, and we will clarify the novelty in the revised manuscript by including the following explanation.
>
> We use a standard white-box PGD attack on the VLM’s vision input, but before each attack we pre-fill the model’s context window with a different amount of content. This allows us to test the effect of different levels of inference compute – i.e., different amounts of pre-filling – on the difficulty of the attack.
>
> This contribution is important because naively letting the model reason freely (dynamically changing the context window’s contents) *during* an attack could destroy the progress made by the PGD attacker: an attack that is effective with one model context may be completely benign when the model context is changed. In Zaremba et al. (2025), this only problem did not arise because only static black box attacks were used: the input image does not change in black-box attacks, which allows measurement of the effectiveness of reasoning against *a constant input attack* as a function of inference compute.
>
> Thus, we provide a white-box attack experiment that tests for the first time (to our knowledge) how inference compute scaling affects PGD attack success. Notably, we also test models using the same black-box setting as Zaremba et al. (2025). Crucially, we observe support for the RICH (our core hypothesis) in both settings.
>
> > Ln311: While the insights are interesting to me, it is somehow not explained in the paper how close this could be helpful for real-world defenses? (practability)
>
> We will revise our submission to clarify the real-world practicality as follows.
>
> Our findings have several practical implications. First, they show that using robust base models is critical—reasoning only improves robustness when built on stable representations. Second, we demonstrate that lightweight, post-hoc adversarial finetuning (e.g., FARE-LLaVA) is sufficient to enhance the robustness benefits of reasoning without costly retraining. Third, our results highlight inference-time reasoning as a viable defense mechanism—when paired with robustness. Finally, we challenge the common belief that scaling compute alone improves robustness, suggesting that resources should instead be allocated toward robust training and finetuning strategies. More broadly, robust representations introduce a distinct and underutilized axis for improving performance—complementary to inference compute and large-scale pretraining—by enabling more effective generalization under adversarial and out-of-distribution conditions.
>
>
> # Joint Response for All Readers
>
> We are glad to know reviewers were “very intrigued” (oVEA) by our work, finding it made “an important contribution” (mxf9) to an “important problem” (ZqwH) with “practical significance” (VKwn). To our knowledge, we are the first to replicate a phenomenon previously only observed in proprietary models (Zaremba et al., 2025). Excitingly, our results are consistent with prior work: obtaining robustness from inference compute is indeed a promising direction. However, we go beyond replicating prior work, focusing on the relatively small improvements previously obtained with multimodal reasoning as a defense against adversaries. We propose the Robustness from Inference Compute Hypothesis (RICH), which predicts when scaling inference compute in multimodal settings will provide robustness benefits. Our submission’s experiments, and new experiments we added for reviewers like visual prompt injection defense, consistently support the RICH.

---

> > ### Comment · Reviewer_mxf9 · 2025-08-04
> >
> > Dear Authors,
> >
> > Thank you for your detailed responses to my review. I appreciate your efforts to clarify several points, and I’m glad to see that you’ve expanded the experiments and incorporated additional details.
> >
> > PGD Novelty: Your explanation of how PGD interacts with context windows to scale inference compute is helpful and clarifies the novelty in your approach. It strengthens your argument.
> >
> > TTA: I’m glad you plan to discuss Test-Time Adaptation (TTA) in future work, as I think it could provide valuable context. It would be interesting to see how it could be related to your work on adversarial vulnerabilities.
> >
> > Real-world Practicality: I found your discussion of robust base models and post-hoc fine-tuning to be insightful. I believe these are crucial for real-world applications, and I’d love to see more concrete examples in the future to further validate the practical implications.
> >
> > Model Size: The inclusion of larger models in your experiments is a step in the right direction. I still believe a broader set of models could help generalize your findings, but I understand the limitations.
> >
> > Overall, your revisions address my concerns, and I look forward to seeing the final version of the paper.

---

> > > ### Author Response · Authors · 2025-08-06
> > >
> > > We are thankful for your helpful comments/questions and engagement throughout the review period. You have helped strengthen this work, and we are excited to share the revised version.
> > >
> > > Thank you,
> > >
> > > Authors of 24976

---

### Official Review · Reviewer_ZqwH · 2025-06-30

**Clarity:** 3
**Significance:** 3
**Originality:** 3
**Rating:** 4
**Confidence:** 3

**Summary:**

This paper proposes the Robustness from Inference Compute Hypothesis (RICH), which suggests that increasing inference-time compute through long reasoning improves model robustness most effectively when attacks are forced into in-distribution regimes that the model can understand. The authors conduct experiments with vision-language models of varying robustness levels under both white-box and black-box attacks. They demonstrate that models with higher baseline robustness gain disproportionately more benefits from additional inference-time compute, while non-robust models see little improvement. The findings indicate that inference-time compute and train-time defenses work synergistically rather than additively.

**Questions:**

1. How would the proposed hypothesis generalize to other types of attacks beyond vision attacks?

2. What are the specific characteristics of "in-distribution" attacks that make them more amenable to reasoning-based defenses?

3. How does the computational cost of increased inference-time compute compare to other robustness approaches?

4. Could there be alternative explanations for why robust models benefit more from increased compute beyond the distribution shift hypothesis?

**Ethical Concerns:**

["NO or VERY MINOR ethics concerns only"]

**Final Justification:**

This paper proposes an interesting hypothesis, which suggests that increasing inference-time compute through long reasoning improves model robustness most effectively when attacks are forced into in-distribution regimes that the model can understand. The experiments results, including black-box and white-box, provide clear evidence supporting their hypothesis.

My reason for not giving a higher score is that this paper is purely experimental without theoretical support, which the authors seem to acknowledge in their rebuttal. The experimental method is also not novel (PGD attack). However, considering this is an interesting observation and hypothesis, I raised my score to 4.

**Limitations:**

yes

**Paper Formatting Concerns:**

No paper formatting concerns

**Quality:**

2

**Strengths And Weaknesses:**

**Strengths**
1. The paper addresses an important problem regarding the effectiveness of inference-time compute scaling as a defense mechanism against adversarial attacks.

2. The methodology is comprehensive, testing the hypothesis through multiple experiments including both white-box and black-box attacks across different models.

3. The paper provides clear evidence supporting their hypothesis through systematic experiments with varying levels of model robustness and inference compute.

**Weaknesses**
1. The theoretical foundation for why inference-time compute should work better with more robust models lacks depth. The paper mainly relies on empirical observations rather than providing a rigorous theoretical analysis.

2. The paper proposes an explanatory hypothesis but does not develop specific improvement methods or new defense strategies based on this hypothesis. The findings remain largely theoretical without practical defensive solutions.

3. The experimental design is relatively straightforward. The experiments primarily validate the hypothesis through increasing computation and comparing models with different robustness levels, without exploring more sophisticated experimental approaches.

4. The experiments are limited to relatively small-scale models as acknowledged in the limitations section. This raises questions about whether the findings would hold true for larger, more sophisticated models.

5. The paper lacks ablation studies to isolate the impact of different components of their approach, making it difficult to understand which aspects are most crucial for the observed improvements.

---

> ### Author Rebuttal · Authors · 2025-07-31
>
> We thank the reviewer for their thoughtful feedback. We would like to reemphasize the broader impact and practical implications of our work: it challenges the prevailing assumption that scaling alone yields robustness, and instead highlights robust representations as a key pathway for improving performance under adversarial conditions. To strengthen our case, we have added new experiments showing: (a) the RICH hypothesis holds for larger open-weight models (LLaMA 3.2 Vision 90B and Qwen2 VL 72B), (b) stronger chain-of-thought prompting significantly improves performance in the adversarial setting for robust models, and (c) RICH applies to semantic adversarial attacks that perform visual prompt injections (see response to Reviewer oVEA). Together, these additions provide further empirical support for RICH and reinforce its practical relevance to building safer, more robust systems. We hope you will champion acceptance of the paper, and we will sincerely appreciate your engagement if there are further questions.
>
> > The paper proposes an explanatory hypothesis but does not develop specific improvement methods or new defense strategies based on this hypothesis. The findings remain largely theoretical without practical defensive solutions.
>
> While our paper introduces an explanatory hypothesis, it also offers practical implications grounded in empirical results. Our findings push back against the common misconception that scaling alone solves robustness. We show that increasing model size or inference compute, without addressing representation robustness, offers limited benefit under adversarial conditions. In contrast, using robust base models enhances inference-time compute’s ability to defend against attacks, improving performance even in challenging settings. We also demonstrate that lightweight, post-hoc adversarial finetuning (e.g., FARE-LLaVA from the Robust CLIP paper) can obtain this effect, which has notable practical relevance. Thus, our findings offer easy-to-use, actionable pathways for improving robustness in deployed systems, not just theoretical insights. We will emphasize this practical dimension more clearly in our revision—thank you for the helpful suggestion.
>
> > The experiments primarily validate the hypothesis through increasing computation and comparing models with different robustness levels, without exploring more sophisticated experimental approaches.
>
> Please let us know if you have ideas for more sophisticated experimental approaches. Our experiments already greatly augmented those considered by Zaremba et al. (2025). In fact, we developed a novel white-box attack that is (to our knowledge) the first white-box attack that has been applied to test inference-time compute as a defense – the adversarial attacks in Zaremba et al. (2025) only include the black-box attacks in the Attack Bard dataset, which we also studied.
>
> > The experiments are limited to relatively small-scale models as acknowledged in the limitations section. This raises questions about whether the findings would hold true for larger, more sophisticated models.
>
> Our revision addresses this by adding new models that are larger than those we explored. As shown in Table R1 below, LLama-3.2-Vision-High-90B and Qwen-2.5-VL-72B-Instruct do not exhibit substantial, statistically significant gains from CoT on attacked data despite doing so on clean data. This supports the RICH, consistent with our submission’s Table 2 and the updated version of Table 2 below (Table R2).
>
> Our revision will mention that future work can adversarially finetune these models to investigate whether the increased robustness from finetuning improves the robustness benefits these models obtain from reasoning.
>
> **Table R1** Attack Bard Black Box 1000-way Classification On Large Models
> | Evaluation | Model | No CoT | CoT | CoT Improvement at 0.01 Significance Level? (P-Value)|
> |----|-------|--------|-----|-----|
> | Clean | LLama-3.2-Vision-High-90B | 63.5 | 68.5 | **No** (1.921e-2) |
> | Clean | Qwen-2.5-VL-72B-Instruct | 57.0 | 67.5 | **Yes** (5.557e-4) |
> | Adv | LLama-3.2-Vision-High-90B | 27.0 | 27.5 | **No** (7.905e-1) |
> | Adv | Qwen-2.5-VL-72B-Instruct | 13.0 | 18.0 | **No** (1.273e-2) |
>
>
> **Table R2** Attack Bard Black Box 30-Way Classification on LLaVA Family
> | Evaluation | Model | No CoT | CoT | CoT Improvement at 0.01 Significance Level? (P-Value) |
> |-----|---|--|---|--|
> | Clean | LLaVA-v1.5 | 69.5 | 82.0 | **Yes** (1.376e-4) |
> | Clean | FARE-LLaVA-v1.5 | 61.5 | 71.0 | **Yes** (9.414e-4) |
> | Clean | Delta2LLaVA-v1.5 | 62.0 | 72.5 | **Yes** (3.919e-3) |
> | Adv | LLaVA-v1.5 | 38.0 | 44.5 | **No** (4.233e-2) |
> | Adv | FARE-LLaVA-v1.5 | 56.0 | 65.5 | **Yes** (4.621e-3) |
> | Adv | Delta2LLaVA-v1.5 | 62.0 | 73.0 | **Yes** (4.509e-3) |
>
> > The paper lacks ablation studies to isolate the impact of different components of their approach, making it difficult to understand which aspects are most crucial for the observed improvements.
>
> Could you please clarify the different approach components that you’re referring to? Our understanding of our approach is that we induced increases in inference time compute in various ways (e.g. through CoT and through a novel text-repetition technique), and we tested whether these increases led to robustness improvements. We consistently found that, regardless of how inference compute was increased, robustness only improved significantly when the base model was already robust.
>
> > How would the proposed hypothesis generalize to other types of attacks beyond vision attacks?
>
> We discuss this in lines 461-467 of our submission, in the appendix, and we will move this discussion into the main text if our paper is accepted (which grants an additional page). This discussion is paraphrased as follows: Our hypothesis suggests that other attacks, like text attacks, may be resisted by increased compute due to the mechanism outlined by the RICH. In particular, since larger LLMs tend to be more robust even without adversarial training (Howe et al., 2024), the large-scale LLMs explored in Zaremba et al. (2025) may have been able to resist text attacks via inference compute through the RICH mechanism. We leave more targeted experiments of this phenomenon in the text modality to future work.
>
> >  [Request for theoretical analysis.] What are the specific characteristics of "in-distribution" attacks that make them more amenable to reasoning-based defenses?
>
> We would like to emphasize that our primary contribution is empirical, and these findings have standalone value. The link between inference-time compute and robustness across attack types is consistent, reproducible, and practically valuable. Publishing such empirical insights provides a foundation for the community to build on, even if a full theory is not yet in place.
>
> Our revision will suggest modeling perception noise or errors in the DS3 framework (Ellis-Mohr et al., 2025) as a potential way to study the impact of robust perception with inference-time compute. This potential future work warrants a separate investigation due to its complexity and out-of-scope nature.
>
> Regarding the reviewer’s question on in-distribution attacks: our hypothesis is that these attacks often exploit superficial correlations that reside on the data manifold. Reasoning-time defenses can better detect and override such patterns by leveraging more semantically aligned representations in robust models. In contrast, out-of-distribution attacks may operate off-manifold, demanding fundamentally different mechanisms—something we are actively investigating.
>
> > How does the computational cost of increased inference-time compute compare to other robustness approaches?
>
> This is a great question, and our revision will include the following discussion to address this. Addressing robustness by increasing inference compute introduces new computational costs. While our work provides evidence for a hypothesis that guides efficient usage of inference compute in adversarial settings, we note that alternative methods (e.g., use of a diffusion-based purification model, Nie et al., 2022) could be more efficient paths to robustness in some circumstances. However, in settings like our white box attack on the red ball image, increasing inference compute is an inexpensive defense. The base instruction is 35 tokens and each inference compute level (1, 2, 3) adds an additional 8 tokens.
>
> > Could there be alternative explanations for why robust models benefit more from increased compute beyond the distribution shift hypothesis?
>
> We would be happy to add to our revision alternative explanations, but our tests of alternative hypotheses found no support. For example, our submission suggested an alternative explanation of our results: optimizing a PGD attack against *any* model may become more difficult as inference compute (context window size) grows, but our experiments found that larger context window sizes did not raise the number of optimization steps required by the PGD attacker – unless the model being attacked had been trained to be robust. We then considered another alternative hypothesis: optimizing a PGD attack against *adversarially trained* models becomes more difficult as inference compute (context window size) grows. However, our experiments again found this explanation to be insufficient: we found that inference compute could make PGD attacks more difficult in models that had not been adversarially trained – if the attacker’s budget was smaller (epsilon = 16). In this case, we did indeed find that (even for non-robust models), inference compute has robustness benefits. In summary, despite considering alternatives, we only found evidence for the idea that inference compute provides benefits when the attacked data is near enough to the attacked model’s training distribution.

---

> > ### Comment · Reviewer_ZqwH · 2025-08-05
> >
> > Thank you for the authors' detailed responses, which have partially addressed my concerns.
> >
> > I still have two questions:
> >
> > 1.Since the white-box PGD attack is optimized on the specific prompt in Figure 4, have you tested if these adversarial images remain effective under different prompts? Should this be considered when evaluating the attack success rate?
> >
> > 2. It would be helpful to specify the settings used in Figure 1 to distinguish it from Figure 5, as these two figures look similar.

---

> > > ### Author Response · Authors · 2025-08-05
> > >
> > > Thank you for your continued, helpful feedback. Addressing these comments is improving the revised manuscript's clarity/presentation. Please let us know if you have any remaining questions/concerns.
> > >
> > > > 1. Since the white-box PGD attack is optimized on the specific prompt in Figure 4, have you tested if these adversarial images remain effective under different prompts? Should this be considered when evaluating the attack success rate?
> > >
> > > We did test the effects of different prompts, and our revision will carefully explain how and when we did this as follows. Notably, we first emphasize that our core finding held throughout all experiment settings: more robust models consistently benefitted more from increased inference compute on adversarial data.
> > >
> > > For the black-box setting, our submission and revision utilize the image attacks from the Attack-Bard dataset, which are known to transfer across prompts (Dong et al., 2023); i.e., they are “universal”. When generating the new black-box results for the rebuttal, given in Tables R1 and R2 above, we tried several distinct prompts. Across these different prompts, we obtained consistent support for our core hypothesis: regardless of the specific prompt, robust models benefitted most on adversarial data from increased inference compute.
> > > - Please note that we obtained results with different prompts during the response period but only shared results from one prompt (used to create the data in Tables R1 and R2). While additional results are no longer allowed to be shared with reviewers, we will include results with other prompts in our revision’s appendix to show that the findings in Tables R1 and R2 are not sensitive to prompt specifics.
> > >
> > > For the white-box setting, our submission explored the prompt shown in Figure 4 as well as 2 variants (attacking color and texture rather than shape). However, in all white-box tests, we did not attempt to create “universal” image attacks, which requires optimizing an image to work under a variety of model prompts/contexts (Bailey et al., 2024; Schaeffer et al., 2024). Instead, our approach created a unique PGD attack for each unique model context, which is a far less constrained problem as the PGD attack needs to work only for a single prompt. Moreover, we evaluated robustness only on the prompt used to construct the PGD attack, which measures a worst-case sense of robustness (i.e., robustness against a strong adversary). Importantly, our white-box results mirrored our black-box results: whether attacks are stronger (white-box) or weaker (black-box), and regardless of the prompt specifics, we find that inference compute adds the most robustness to models that are already robust.
> > > - Please note that, in addition to using three prompt variations in our original submission, our revised white-box experiments now include more images, as our response to Reviewer **oVEA** clarifies in its discussion and its Table R1.
> > >
> > >
> > >
> > > > 2. It would be helpful to specify the settings used in Figure 1 to distinguish it from Figure 5, as these two figures look similar.
> > >
> > > We agree and will clarify this! In the Figure 5 caption (and line 226), our submission specifies that Figure 5 shows the PGD attacker’s loss when the attacker’s budget $\epsilon$ is large ($\epsilon=$64/255) – only the most robust model (Delta2LLaVA) obtains robustness benefits from inference compute scaling in this case. However, Figure 1’s caption is less clear.
> > >
> > > Our revision will clarify that Figure 1 shows the smallest tested $\epsilon$ that can produce a successful attack: this value is $\epsilon=$64/255 for Delta2LLaVA and $\epsilon=$16/255 for the other models. Our revision will also clarify that Figure 6 summarizes Figure 1’s and Figure 5’s key results, giving the optimization step on which the PGD attacker succeeded for both attack budgets as a function of inference compute. Delta2LLaVA is the most robust model and obtains the clearest benefits from inference compute (and cannot even be attacked at $\epsilon=$16/255), while the moderately robust and non-robust models exhibit some benefit from inference compute provided that the attack budget is small ($\epsilon=$16/255), consistent with the idea that inference compute aids robustness more as the attacked data moves closer to the training distribution (i.e., consistent with the robustness from inference compute hypothesis, RICH).

---

> > > > ### Comment · Reviewer_ZqwH · 2025-08-05
> > > >
> > > > Thank you for the response. I now understand your approach in the white-box tests. I have no more concerns.

---

> > > > > ### Author Response · Authors · 2025-08-06
> > > > >
> > > > > Thank you for your thorough engagement with our work. We are pleased to hear that you have no more concerns. Given that we have successfully addressed all of your questions, we hope you will accordingly recommend our work’s acceptance.
> > > > >
> > > > > We are excited to share our clearer revision, which contains augmented experiments as well as a new experiment on visual prompt injection attacks in an open-ended setting (Table R3 in our response to Reviewer **VKwn**), all of which support our core hypothesis.
> > > > >
> > > > > Thank you again for your valuable feedback that has helped strengthen our paper.
> > > > >
> > > > > Authors of 24976

---

> > > > > > ### Comment · Reviewer_ZqwH · 2025-08-08
> > > > > >
> > > > > > I have raised my score to 4. Good luck.

---

### Official Review · Reviewer_VKwn · 2025-07-02

**Clarity:** 3
**Significance:** 3
**Originality:** 2
**Rating:** 3
**Confidence:** 2

**Summary:**

This paper investigates when and why increasing inference-time compute—such as adding longer prompts or Chain-of-Thought reasoning-can improve robustness against adversarial attacks. The authors introduce the RICH (Robustness-from-Inference-Compute Hypothesis), which posits that extra compute is only beneficial when the model has already been trained to be somewhat robust. Through white-box and black-box adversarial evaluations on three vision-language models of increasing robustness, they show that only the more robust models experience significant gains from inference-time scaling. The paper provides both a theoretical framing and empirical evidence that robustness and inference compute interact synergistically rather than additively, offering clear guidance for how to prioritize robustness efforts.

**Questions:**

- All experiments use 7 B LLaVA derivatives on image-classification prompts. Does RICH hold for larger VLMs or open-ended chat?
- Could the authors provide more precise measurements (e.g., FLOPs, runtime) for the various inference-compute regimes, to better compare across models and to real-world settings?

**Ethical Concerns:**

["NO or VERY MINOR ethics concerns only"]

**Final Justification:**

I will not change my score. While the authors engaged constructively in the discussion and provided many additional results, the original work felt relatively narrow in scope and the experimental coverage limited. The new results from the rebuttal phase add substantial value; however, their integration into the paper, along with the accompanying statistical analysis, may significantly alter the reported findings and conclusions. A thorough re-evaluation of the work under these new results will be important for the final version.

**Limitations:**

Yes.

**Paper Formatting Concerns:**

I did not observe any significant deviations from the NeurIPS 2025 formatting guidelines.

**Quality:**

2

**Strengths And Weaknesses:**

Strengths
- Clear, accessible writing: The prose is concise and logically structured, making technical sections easy to follow even on a first read.
- Reader-friendly Q&A framing: Throughout the paper the authors pose explicit questions (e.g., “Does compute help all models equally?” around lines 231 and 253) and answer them immediately, which highlights the key take-aways and guides the reader through the experimental narrative.
- Methodological rigor: Three vision-language models representing a spectrum of train-time robustness are evaluated under both a new white-box PGD attack and a black-box transfer set, with systematic variation of inference-time compute.
- Original contribution: The work formalises the RICH hypothesis and introduces the first white-box attack specifically designed to probe inference-compute defences in multimodal settings.
- Practical significance: Results show a “rich-get-richer” effect—robust models gain up to 5× more resistance when compute scales—offering concrete guidance to practitioners on when extra inference budget is worthwhile.

Weaknesses
- Limited scope of models and tasks: All experiments are confined to 7 B-class LLaVA derivatives and image-classification prompts; it remains unclear whether RICH generalises to larger frontier models or open-ended chat scenarios.
- Compute overhead unreported in the main text: High-compute settings allow very long “thinking” traces yet provide no latency, throughput, or cost analysis, leaving real-world viability uncertain.
- Small adversarial corpus: The black-box Attack-Bard set contains only 200 images, which may limit statistical power and diversity.
- Reproducibility gaps: Critical implementation details (exact prompt templates, PGD step sizes, per-ε results) are pushed to appendices, making replication from the main paper alone difficult.
- Incremental originality of idea: While applying inference-compute scaling to vision is novel, the core intuition—that extra compute can improve robustness—extends earlier text-only studies rather than introducing a fundamentally new mechanism.

---

> ### Author Rebuttal · Authors · 2025-07-31
>
> Thank you for your thoughtful review!  We are happy you identified that “robustness and inference compute interact synergistically rather than additively, offering clear guidance for how to prioritize robustness efforts.” Indeed, our findings challenge the current notion that scaling alone solves robustness. Our work in this new area suggests a guiding principle for future adversarial defenses: additional inference compute is wasted without representation robustness. Regarding the scope of our experiments, please note that we have expanded our results with frontier open-weight models and a new open-ended setting that bolsters the generality of evidence for the Robustness-from-Inference-Compute Hypothesis (RICH).
>
> > Limited scope of models and tasks: All experiments are confined to 7 B-class LLaVA derivatives and image-classification prompts; it remains unclear whether RICH generalises to larger frontier models or open-ended chat scenarios.
>
> As we discuss below, our revision will broaden our analysis to include frontier multimodal open-weight models (Llama 3.2 Vision 90B and Qwen 2 VL 72B), extending our black box transfer experiments. Additionally, we study the RICH in the context of visual prompt injection attacks that mirror **open-ended** real-world settings.
>
> First, we add Llama 3.2 Vision 90B and Qwen 2 VL 72B to the Attack-Bard experiments (shown in Table R1 below). Given that these models have not been adversarially trained, their robustness to adversarial image attacks is limited. The Robustness from Inference Compute Hypothesis (RICH) – which predicts that reasoning on non-robust data representations is less beneficial than reasoning on robust representations (i.e. from data that is more in-distribution) – would therefore be supported if these models had less improvement from inference compute on adversarial data than on clean data. Comparing the relative accuracy of chain-of-thought to no reasoning, we find that raising inference compute clearly helps larger models on clean data. However, raising inference compute provides no statistically significant (p-value < 0.01) benefit on adversarial data within each model family.
>
> In contrast, when we use a robust base model in the same setting (Table 2 and its updated version Table R2, below), raising inference compute improves both clean and adversarial data performance, with adversarial data performance benefitting the most. This underscores our key finding: scaling model size or inference compute alone won’t address robustness. Raising awareness of this limitation can avoid potentially inefficient (or ineffective) scaling.
>
> **Table R1**: Attack-Bard Black Box 1000-Way Classification on Frontier Models
> | Evaluation | Model | No CoT | CoT | CoT Improvement at 0.01 Significance Level? (P-Value)|
> |------------|-------|--------|-----|--------------------------------------------|
> | Clean | LLama-3.2-Vision-High-90B | 63.5 | 68.5 | **No** (1.921e-2) |
> | Clean | Qwen-2.5-VL-72B-Instruct | 57.0 | 67.5 | **Yes** (5.557e-4) |
> | Adversarial | LLama-3.2-Vision-High-90B | 27.0 | 27.5 | **No** (7.905e-1) |
> | Adversarial | Qwen-2.5-VL-72B-Instruct | 13.0 | 18.0 | **No** (1.273e-2) |
>
> **Table R2**: Attack-Bard Black Box 30-Way Classification on LLaVA Family
> | Evaluation | Model | No CoT | CoT | CoT Improvement at 0.01 Significance Level? (P-Value) |
> |------------|-------|--------|-----|------|
> | Clean | LLaVA-v1.5 | 69.5 | 82.0 | **Yes** (1.376e-4) |
> | Clean | FARE-LLaVA-v1.5 | 61.5 | 71.0 | **Yes** (9.414e-4) |
> | Clean | Delta2LLaVA-v1.5 | 62.0 | 72.5 | **Yes** (3.919e-3) |
> | Adversarial | LLaVA-v1.5 | 38.0 | 44.5 | **No** (4.233e-2) |
> | Adversarial | FARE-LLaVA-v1.5 | 56.0 | 65.5 | **Yes** (4.621e-3) |
> | Adversarial | Delta2LLaVA-v1.5 | 62.0 | 73.0 | **Yes** (4.509e-3) |
>
> Second, we find evidence for the RICH in a setting other than image classification using non-robust and robust models from the LLaVA series. We inserted a visual prompt injection into a clean Attack-Bard image, then we performed a white-box PGD attack that encouraged the model to repeat the visually injected text instead of describing the image as asked. When we scaled the inference compute, only robust models benefitted, as shown by the increase in the loss of the PGD attacker below. This extends the generality of our findings and broadens RICH support to additional attack methods.
>
> **Table R3**: Only robust LLaVA models obtain added robustness (measured with PGD attacker’s loss) from inference compute scaling when faced with visual prompt injections.
>
> | Model | Base Model Robustness | Inference Compute | Step 100 Attacker Loss (&uarr;) | Step 200 Attacker Loss (&uarr;) | Step 300 Attacker Loss (&uarr;) | Inference Compute Robustness Effect |
> | --- | --- | --- | --- | --- | --- | --- |
> | Delta2LLaVA-v1.5 | High | Low | 13.5 (0.0) | 12.7 (0.1) | 12.4 (0.0) | -- |
> | Delta2LLaVA-v1.5 | High | High | 21.2 (0.0) | 21.1 (0.0) | 21.1 (0.0) | Positive |
> | FARE-LLaVA-v1.5 | Medium | Low | 7.5 (0.4) | 7.0 (0.0) | 7.0 (0.5) | -- |
> | FARE-LLaVA-v1.5 | Medium | High | 9.3 (1.1) | 7.4 (0.7) | 7.2 (0.3) | Neutral |
> | LLaVA-v1.5 | Low | Low | 6.4 (1.4) | 2.6 (2.9) | 2.0 (2.6) | -- |
> | LLaVA-v1.5 | Low | High | 2.9 (0.8) | Attack Success | Attack Success | Negative |
>
> > Compute overhead unreported in the main text: High-compute settings allow very long “thinking” traces yet provide no latency, throughput, or cost analysis, leaving real-world viability uncertain.
>
> Our revision will include the following discussion on the cost implications of increasing robustness via inference compute. Addressing robustness by increasing inference compute introduces new computational costs. While our work provides evidence for a hypothesis that guides efficient usage of inference compute in adversarial settings, we note that alternative methods (e.g., use of a diffusion-based purification model, Nie et al., 2022) could be more efficient paths to robustness in some circumstances. In settings like our white box attack on the red ball image, increasing inference compute is a simple and computationally inexpensive defense. The base instruction is 35 tokens and each inference compute level (1, 2, 3) adds an additional 8 tokens. This suggests that robust base models can have their defenses bolstered cheaply in some scenarios.
>
>
> > Small adversarial corpus: The black-box Attack-Bard set contains only 200 images, which may limit statistical power and diversity.
>
> Please note that the experiments in Zaremba et al. (2025) only use Attack-Bard for their adversarial multimodal experiments. We use the same dataset to follow their experiment methodology. Additionally, we go beyond their experiments by introducing new data and methodologies (e.g. our novel white-box attack setup).
>
> However, our revision will address the concern about statistical power. We have updated our reporting of the black-box experiment results in Table 2 such that p-values are now visible (we also modified the CoT prompt to improve its effectiveness on clean data). The updated table R2 is above. Despite only using 200 images (like Zaremba et al., 2025), we find statistically significant benefits (p<0.01) of CoT on the Attack-Bard task when the model is robust or the data is clean – if the data is attacked/unclean and the model is not robust, then the benefit of CoT is not statistically significant. These findings are consistent with and statistically support the RICH.
>
> Additionally, please note that our submission’s white-box experiments reported standard errors to provide statistical support for our claims in that setting (see Table 1).
>
>
> > Reproducibility gaps: Critical implementation details (exact prompt templates, PGD step sizes, per-ε results) are pushed to appendices, making replication from the main paper alone difficult.
>
> We appreciate the reviewer’s emphasis on reproducibility and agree that critical implementation details are essential for replication. We would like to clarify that all such details (e.g., exact prompt templates, PGD step sizes, per-ε results) are fully included in the appendix and are explicitly referenced in the main text. While page limitations currently preclude their inclusion in the main body, we plan to move them into the main text upon acceptance, when an extra page becomes available.
>
> >Incremental originality of idea: While applying inference-compute scaling to vision is novel, the core intuition—that extra compute can improve robustness—extends earlier text-only studies rather than introducing a fundamentally new mechanism.
>
> We disagree about how original our work is and the core intuition: we do not simply extend earlier studies to the vision domain and show “that extra compute can improve robustness”. Instead, we introduce a novel hypothesis and explanatory mechanism that indicates when inference compute will benefit robustness. Further, we show that our predictions are consistent with experiments: models predicted to not significantly benefit from inference compute do not, and models predicted to significantly benefit from inference compute do. In summary, beyond overcoming the limited benefits to multimodal robustness achieved via inference compute by Zaremba et al. (2025), we provide an explanation for why these benefits are achieved and leverage it to obtain our superior results.

---

> > ### Comment · Area_Chair_E4ow · 2025-08-05
> >
> > Dear reviewer,
> >
> > The discussion phase is soon coming to and end. It will be great if you could go over the rebuttal and discuss with the authors if you still have outstanding concerns. Thank you for being part of the review process.
> >
> > Regards,
> >
> > Area Chair

---

> > > ### Author Response · Authors · 2025-08-06
> > >
> > > Hi Reviewer VKwn,
> > >
> > > Thanks again for your review. We’ve posted a comprehensive rebuttal that we believe addresses all of the points you raised. If you have additional questions or things to discuss, please let us know. If we have addressed all of your concerns, we’d appreciate your according support of our submission's acceptance.
> > >
> > > Thank you,
> > >
> > > Authors of 24976

---

> > ### Comment · Reviewer_VKwn · 2025-08-09
> >
> > Thank you for the strong rebuttal and the additional experiments, as well as the useful statistical analysis, which will help in drawing stronger conclusions. I understand the rationale for following the Zaremba et al. methodology; however, such approaches should continue to evolve and be evaluated more thoroughly. The new results and analyses may necessitate substantial changes to the manuscript to ensure the findings are fully reflected and clearly presented.

---

### Comment · Area_Chair_E4ow · 2025-08-01

Dear reviewers,

Please read the rebuttal on time and put forward further comments (if any) to facilitate the authors' response.

Best, AC

---

### Note · Authors · 2025-08-12

Thank you again, reviewers and AC. Below, we address concerns **VKwn** posted at the discussion period’s end.

> Thank you for the strong rebuttal and the additional experiments, as well as the useful statistical analysis

We are glad these helped and hope they raise your score.

> ​​I understand the rationale for following the Zaremba et al. methodology; however, such approaches should continue to evolve and be evaluated more thoroughly

Please note that **our submission already evolved the approach of Zaremba et al. (2025):**

- **Open models:** ​​For the first time, we show open models exhibit the link between inference compute and robustness previously only observed in a single proprietary model.

- **Multiple base model robustness levels and sizes:** We extend evaluation to models of various sizes and robustness levels, clarifying that robustness gains depend predictably on the base model’s robustness.

- **Statistical rigor:** On the Attack-Bard dataset, prior work reported inference compute’s robustness benefit without statistics. *We find that this benefit is only statistically significant for robust base models.*

- **Attack diversity:** While prior work only tested black-box classification attacks, *our submission tested both white-box and black-box attacks*.

Notably, *our revision will further broaden the tests of Zaremba et al. (2025) by adding an experiment with visual prompt injections in open-ended chat.* Critically, across all experiments, we observe support for our “original” (**VKwn**) hypothesis. This hypothesis is predictive of experimental results, and it gives “concrete guidance to practitioners on when extra inference budget is worthwhile” (**VKwn**).

> The new results and analyses may necessitate substantial changes to the manuscript to ensure the findings are fully reflected and clearly presented

To be clear: **our revisions are clarifications and evidence expansions only**. The core hypothesis, methodology, and findings remain exactly as submitted. The new experiments (e.g., visual prompt injections, 90B models) were added to address reviewer requests for broader scope. These additions *strengthen* support for our original claim without changing it; e.g., our revision adds a new table column to show statistical test results. Thus, our revision does not risk losing the "methodological rigor" and "practical significance" that **VKwn** praised our submission for – it instead makes gains on these fronts through minor edits.

---

### Decision · Program_Chairs · 2025-09-17

**Decision:**

Reject

**Comment:**

This paper tackles an interesting question—when and why adding inference‑time compute improves robustness—and articulates the RICH hypothesis with empirical evidence across black‑/white‑box settings.  Most reviewers are positive after the rebuttal.

However, in a highly selective year, we must prioritize fully consolidated manuscripts. The core evidence remains somewhat narrow (heavy reliance on a small black‑box set), several key extensions emerged during discussion but are not yet integrated and systematized in the main paper, and important pieces are still missing: a thorough cost–benefit analysis (FLOPs/latency/energy), broader datasets/models/tasks with ablations (e.g., disentangling inference compute vs. context length vs. attack budget), and a tighter theoretical framing. Given these factors and the overall acceptance rate, after discussion with SAC via email, I recommend rejection and strongly encourage the authors to address the issues for the submission to subsequent conferences.